# Variational Search Distributions

**Dan Steinberg, Rafael Oliveira, Cheng Soon Ong & Edwin V. Bonilla**
CSIRO's Data61, Australia
{dan.steinberg, rafael.dossantosdeoliveira, cheng-soon.ong, edwin.bonilla}@data61.csiro.au

## Abstract

We develop variational search distributions (VSD), a method for finding and generating discrete, combinatorial designs of a rare desired class in a batch sequential manner with a fixed experimental budget. We formalize the requirements and desiderata for active generation and formulate a solution via variational inference. In particular, VSD uses off-the-shelf gradient based optimization routines, can learn powerful generative models for designs, and can take advantage of scalable predictive models. We empirically demonstrate that VSD can outperform existing baseline methods on a set of real sequence-design problems in various biological systems.

## 1 Introduction

We consider a variant of the *active search* problem [15, 22, 51], where we wish to find members (designs) of a rare desired class in a batch sequential manner with a fixed experimental budget. We call sequential active learning of a *generative* model of these designs **active generation**. Examples of rare designs are compounds that could be useful pharmaceutical drugs, or highly active enzymes for catalyzing chemical reactions. We assume the design space is discrete or partially discrete, high-dimensional, and practically *innumerable*. For example, the number possible configurations of a single protein is $20^{\mathcal{O}(100)}$ [see, e.g., 38].

We are interested in this active generation objective for a variety of reasons. We may wish to study the properties of the "fitness landscape" [33] to gain a better scientific understanding of a phenomenon such as natural evolution. Or, we may not be able to completely specify the constraints and objectives of a task, but we would like to characterize the space of, and generate new feasible designs. For example, we want enzymes that can degrade plastics in an industrial setting, but we may not yet know the exact conditions (e.g. temperature, pH), some of which may be anti-correlated with enzyme catalytic activity. Alternatively, if we know these multiple objectives and constraints, we may only want to generate designs from a Pareto set.

Assuming we can take advantage of a prior distribution over designs, we formulate the search problem as inferring the posterior distribution over rare, desirable designs. Importantly, this posterior can be used for *generating new designs*. Specifically, we use (black-box) variational inference (VI) [35], and so refer to our method as variational search distributions (VSD). Our major contributions are: (1) we formulate the batch active generation objective over a (practically) innumerable discrete design space, (2) we present a variational inference algorithm, VSD, which solves this objective, and (3) we show that VSD performs well empirically. VSD uses off-the-shelf gradient based optimization routines, is able to learn powerful generative models, and can take advantage of scalable predictive models. In our experiments we show that VSD can outperform existing baseline methods on a set of real applications. Finally, we evaluate our approach on the related sequential black-box optimization (BBO) problem, where we want to find the globally optimal design for a specific objective and show competitive performance when compared with state-of-the-art methods.

Workshop on Bayesian Decision-making and Uncertainty, 38th Conference on Neural Information Processing Systems (NeurIPS 2024).

## 2 Problem Formulation

We are given a design space $\mathcal{X}$, which can be discrete or mixed discrete-continuous and high dimensional, and where for each instance that we choose $\mathbf{x} \in \mathcal{X}$, we measure some corresponding property of interest (so-called fitness) $y \in \mathbb{R}$. For example, in our motivating application of DNA/RNA or protein sequences (henceforth referred to as just sequences), $\mathcal{X} = \mathcal{V}^M$ where $\mathcal{V}$ is the sequence vocabulary (e.g., amino acid labels, $|\mathcal{V}| = 20$) and $M$ is the length of the sequence. However, we do not limit the application of our method to sequences. Using this framing, a real world experiment (for example, measuring the activity of an enzyme) can be modeled as an unknown relationship,

$$y = f_{\bullet}(\mathbf{x}) + \epsilon, \tag{1}$$

for some black-box function (e.g. the experiment), $f_{\bullet}$, and measurement error $\epsilon \in \mathbb{R}$, distributed according to $p(\epsilon)$ with $\mathbb{E}_{p(\epsilon)}[\epsilon] = 0$. Instead of modeling the whole space, $\mathcal{X}$, we are only interested in a set of events which we choose based on fitness, $\mathcal{S} \subset \mathcal{X}$. In particular for active generation we wish to learn a generative model, by efficiently querying the black-box function in Equation 1, that only returns samples $\mathbf{x}^{(s)} \in \mathcal{S}$. For the purposes of this work, we define this solution space as the super level-set, $\mathcal{S} := \{\mathbf{x} : y > \tau\}$ for $\tau \in \mathbb{R}$ (e.g., wild-type fitness), and so *our task is to learn the super level-set distribution*, $p(\mathbf{x}|y > \tau)$, *in an active manner*[1]. We contrast this objective to the related objectives of; BBO for the fittest design, $\mathbf{x}^* = \operatorname{argmax}_{\mathbf{x}} f_{\bullet}(\mathbf{x})$, directly estimating the super level-set, $\mathcal{S}$, or the shape of the black-box function for the super level-set, $\mathcal{F} := \{f_{\bullet}(\mathbf{x}) : \mathbf{x} \in \mathcal{S}\}$. We visualize these related objectives in Figure 1. We assume that $\mathcal{S} \subset \mathcal{X}$ are rare events in a high dimensional space, and that we have access to a prior belief, $p(\mathbf{x})$, which helps narrow in on this subset of $\mathcal{X}$. We are given a dataset, $\mathcal{D}_N := \{(y_n, \mathbf{x}_n)\}_{n=1}^N$, which may contain only a few instances of $y_n > \tau$. Given $p(\mathbf{x})$ and $\mathcal{D}_N$ we aim to recommend batches of unique candidates, $\{\mathbf{x}_{bt}\}_{b=1}^B$, for experimental evaluation in a series of rounds, $t \in \{1, \dots, T\}$, where $B = \mathcal{O}(1000)$ and we desire $\mathbf{x}_{bt} \in \mathcal{S}$. Each round, $\mathcal{D}_N$ is augmented with the experimental results of the previous batch, so $N \leftarrow N + B$. Estimating this super level-set distribution of $\mathbf{x}$ is computationally and statistically challenging and, therefore, we cast this as a *variational inference* problem. As we shall see later, our solution allows us to satisfy the following requirements and additional desiderata for our problem.

**Requirements & Desiderata.** *Problem requirements (R) and other desiderata (D).*

**(R1) Rare** *feasible designs, $\mathcal{S}$, are rare events in $\mathcal{X}$ that need to be identified*

**(R2) Sequential** *non-myopic candidate generation, $\mathbf{x} \in \mathcal{S}$, for sequential experimentation*

**(R3) Discrete** *search over (combinatorially) large design spaces, e.g. $\mathbf{x} \in \mathcal{X} = \mathcal{V}^M$*

**(R4) Batch** *generation of up to $\mathcal{O}(1000)$ diverse candidate designs per round*

**(R5) Generative** *models, $\mathbf{x}^{(s)} \sim q(\mathbf{x})$, that are task-specific for fit designs*

**(D1) Guaranteed** *convergence for certain choices of priors, variational distributions and predictive models*

**(D2) Gradient** *based optimization strategies for candidate searching*

**(D3) Scalable** *predictive models that enable high-throughput experiments.*

## 3 Variational Search Distributions

We cast the estimation of $p(\mathbf{x}|y > \tau)$ as a sequential optimization problem using variational inference. To do this we optimize the well-known evidence lower bound (ELBO), $\mathcal{L}_{\text{ELBO}}(\phi) = \mathbb{E}_{q(\mathbf{x}|\phi)}[\log p(y > \tau|\mathbf{x})] - \mathbb{D}_{\text{KL}}[q(\mathbf{x}|\phi)\|p(\mathbf{x}|\mathcal{D}_0)]$, where we assume access to a prior distribution over the space of designs, $p(\mathbf{x}|\mathcal{D}_0)$, that may be informed/pre-trained. Furthermore, we estimate $\log p(y > \tau|\mathbf{x})$ using a surrogate model by recognizing an equivalence between this distribution and the probability of improvement (PI) acquisition function from Bayesian optimization (BO) [25],

$$\log p(y > \tau|\mathbf{x}) \approx \log p(y > \tau|\mathbf{x}, \mathcal{D}_N) = \log \mathbb{E}_{p(y|\mathbf{x},\mathcal{D}_N)}[\mathbb{1}[y > \tau]] = \log \alpha_{PI}(\mathbf{x}, \mathcal{D}_N, \tau). \tag{2}$$

Here $\mathbb{1} : \{\text{false}, \text{true}\} \to \{0, 1\}$ is the indicator function and $p(y|\mathbf{x}, \mathcal{D}_N)$ is typically estimated using the posterior predictive distribution of a Gaussian process (GP) given data $\mathcal{D}_N$. So

---

[1]One could consider other definitions of this solution set, $\mathcal{S}$, for example the Pareto set of non-dominated designs in a multi-objective optimization setting. We leave the formulation of VSD for other $\mathcal{S}$ as future work.

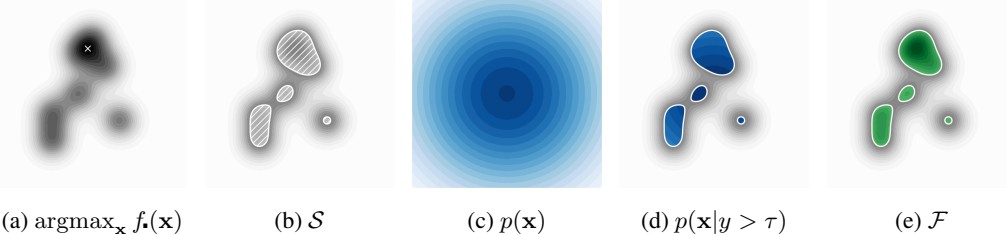

(a) $\mathrm{argmax}_{\mathbf{x}}\, f_\bullet(\mathbf{x})$    (b) $\mathcal{S}$    (c) $p(\mathbf{x})$    (d) $p(\mathbf{x}|y > \tau)$    (e) $\mathcal{F}$

Figure 1: Fitness landscape tasks. (a) $f_\bullet(\mathbf{x})$ and white '×' — the maximum fitness design, $\mathbf{x}^*$. (b) white hatched area — the super-level set of all fit designs, $\mathcal{S}$. (c) prior belief $p(\mathbf{x})$. (d) blue contours — the density of the super-level set, $p(\mathbf{x}|y > \tau)$. (e) the black box function for the super-level set, $\mathcal{F}$. See the text for definitions of these tasks. Our primary goal is to estimate (d).

$p(y > \tau | \mathbf{x}, \mathcal{D}_N) = \Psi((\mu_N(\mathbf{x}) - \tau)/\sigma_N(\mathbf{x}))$, where $\Psi(\cdot)$ is a cumulative standard normal distribution function, and $\mu_N(\mathbf{x})$, $\sigma_N^2(\mathbf{x})$ are the posterior predictive mean and variance, respectively, of the GP. We can now rewrite the ELBO as,

$$\mathcal{L}_{\mathrm{ELBO}}(\phi, \tau, \mathcal{D}_N) = \mathbb{E}_{q(\mathbf{x}|\phi)}[\log \alpha_{PI}(\mathbf{x}, \mathcal{D}_N, \tau)] - \mathbb{D}_{\mathrm{KL}}[q(\mathbf{x}|\phi)\|p(\mathbf{x}|\mathcal{D}_0)]. \tag{3}$$

We refer to our method that optimizes the objective in Equation 3 as VSD, as we are using the variational posterior distribution as a means of searching the space of fit sequences, satisfying (R1), (R2) and (R4). Concretely, we draw a set of sample candidates from our search distribution, (R5), each round,

$$\{\mathbf{x}_{bt}\} \sim \prod_{b=1}^{B} q(\mathbf{x}|\phi_t^*), \quad \text{where} \quad \phi_t^* = \operatorname*{argmax}_{\phi} \mathcal{L}_{\mathrm{ELBO}}(\phi, \tau, \mathcal{D}_N). \tag{4}$$

In general, because of the discrete combinatorial nature of our problem, we cannot readily use the re-parametrization trick to estimate the gradients of the ELBO above. Instead, we use of the score function gradient estimator [30] with standard gradient descent methods (D2),

$$\nabla_\phi \mathcal{L}_{\mathrm{ELBO}}(\phi, \tau, \mathcal{D}_N) = \mathbb{E}_{q(\mathbf{x}|\phi)}\left[\left(\log \alpha_{PI}(\mathbf{x}, \mathcal{D}_N, \tau) - \log \frac{q(\mathbf{x}|\phi)}{p(\mathbf{x}|\mathcal{D}_0)}\right) \nabla_\phi \log q(\mathbf{x}|\phi)\right], \tag{5}$$

where we use Monte-Carlo sampling to approximate this expectation with a suitable variance reduction scheme, such as using a control variate or baseline. We find that the exponentially smoothed average of the ELBO works well in practice, and is the same strategy employed in Daulton et al. [13]. Effectively, VSD implements black-box variational inference [35] for parameter estimation, and despite the high-dimensional nature of $\mathcal{X}$, we find we only need $\mathcal{O}(1000)$ samples to estimate the required expectations for ELBO optimization on problems with $M = \mathcal{O}(100)$, satisfying (R3).

**Class probability estimation**: So far our method indirectly computes the PI acquisition function by transforming the predictions of a GP surrogate model, $p(y|\mathbf{x}, \mathcal{D}_N)$, as in Equation 2. Instead we may choose to follow the reasoning used by Bayesian optimization by density-ratio estimation (BORE) in [47, 32, 40] and directly estimate the quantity we care about, $p(y > \tau | \mathbf{x}, \mathcal{D}_N)$. We do this with class probability estimation (CPE) using $p(z = 1|\mathbf{x}, \mathcal{D}_N) \approx \pi_\theta(\mathbf{x})$, where $z := \mathbb{1}[y > \tau] \in \{0, 1\}$, and $\pi_\theta : \mathcal{X} \to [0, 1]$. We can recover the class probability estimates using a proper scoring rule [16] such as Brier score or log-loss on training data, $\mathcal{D}_N^z = \{(z_n, \mathbf{x}_n)\}_{n=1}^N$, e.g.,

$$\mathcal{L}_{\mathrm{CPE}}(\theta, \mathcal{D}_N^z) := \frac{1}{N} \sum_{n=1}^{N} z_n \log \pi_\theta(\mathbf{x}_n) + (1 - z_n) \log(1 - \pi_\theta(\mathbf{x}_n)). \tag{6}$$

The VSD objective using CPE becomes,

$$\mathcal{L}_{\mathrm{ELBO}}(\phi, \theta, \mathcal{D}_N) = \mathbb{E}_{q(\mathbf{x}|\phi)}[\log \pi_\theta(\mathbf{x})] - \mathbb{D}_{\mathrm{KL}}[q(\mathbf{x}|\phi)\|p(\mathbf{x})], \tag{7}$$

into which we plug $\theta_t^* = \operatorname*{argmax}_\theta \mathcal{L}_{\mathrm{CPE}}(\theta, \mathcal{D}_N^z)$. Using a CPE also opens up the choice of estimators that are more scalable than a GP surrogate, satisfying our last desideratum (D3). This may be crucial if we choose to run more than a few rounds of experiments with $B = \mathcal{O}(1000)$. Additionally, since VSD is a black box method, we can choose to use CPEs that are non-differentiable, such as decision tree ensembles.

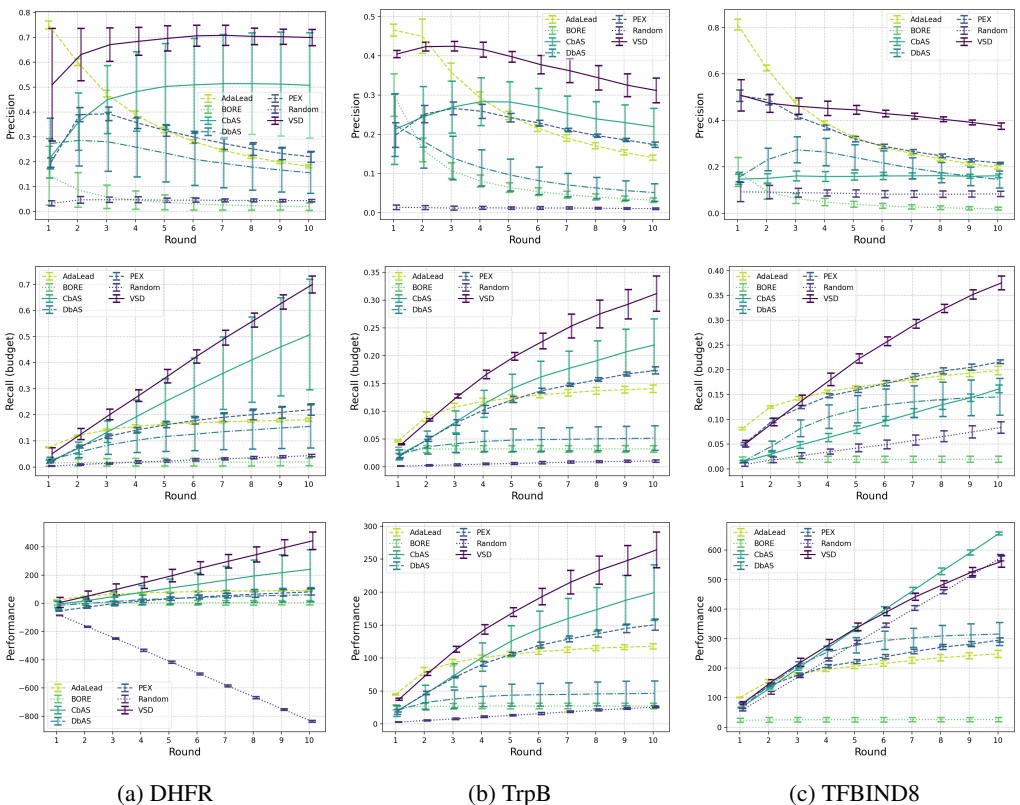

|     |     |     |
| --- | --- | --- |
| (a) DHFR | (b) TrpB | (c) TFBIND8 |

Figure 2: Fitness landscape results. Precision (Equation 13), recall (Equation 14) and performance (Equation 15) – higher is better – for the combinatorially (near) complete datasets, DHFR and TrpB and TFBIND8. The random method is implemented by drawing $B$ samples uniformly.

The complete VSD algorithm is given in Algorithm 1 (Appendix D), in which we have allowed for a threshold function, $\tau_t = f_\tau(\{y : y \in \mathcal{D}_N\}, \gamma_t)$. This function can be used to modify the threshold each round, e.g. following [47], an empirical quantile function $\tau_t = \hat{Q}_y(\gamma_t)$ where $\gamma_t \in (0, 1)$, or a constant $\tau$ in the case of estimating the density of the super-level set.

**Theoretical analysis and related work**: We show in Appendix A that the VSD objective, in fact, generalizes the BO objective, providing a lower bound that is tight iff the prior is a Dirac delta distribution centered at $\mathbf{x}_t^*$. In the sequel [44] we provide convergence guarantees for VSD, satisfying desideratum (D1). In Appendix B we provide a formulation that generalizes several related optimization algorithms (and VSD) including Bayesian optimization with probabilistic reparameterisation (BOPR) [13], design by adaptive sampling (DbAS) [9], conditioning by adaptive sampling (CbAS) [8] and BORE [47]. The key takeaway is that, as seen in Table 2, VSD satisfies all the requirements and desiderata for our problem.

# 4 Experiments

We evaluate our method, VSD, on a number of real-world sequence design tasks involving various biological systems. The corresponding datasets involve $|\mathcal{V}| \in \{4, 20\}$, $8 \le m \le 237$ and $65,000 < |\mathcal{X}| < 20^{237}$. We carry out fitness landscape experiments where we assess the quality of *all* the sequences proposed by the competing algorithms and black-box optimization (BBO) experiments where we evaluate the best performing sequence. We use a mean field variational distribution and independent prior for the fitness landscape experiments, and we also use a long short-term memory (LSTM) and decoder-transformer variational distribution and prior for the higher dimensional BBO experiments. See Appendix C for full details of the experiments, including a description of the evaluation metrics and results on batch diversity.

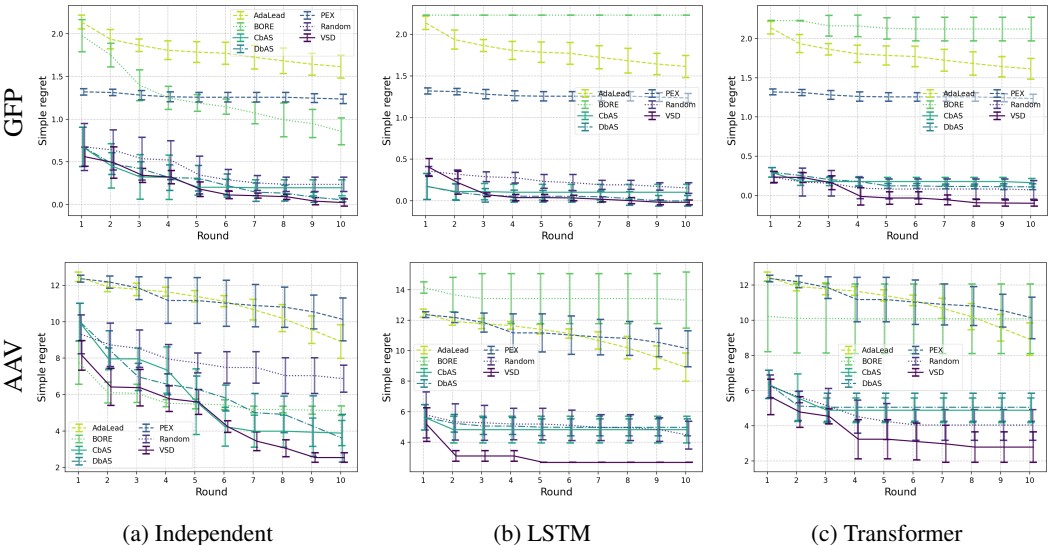

|     | (a) Independent | (b) LSTM | (c) Transformer |
|-----|-----------------|----------|-----------------|

Figure 3: AAV & GFP BBO results. Simple regret (Equation 17) – lower is better – on GFP and AAV with independent and auto-regressive variational distributions. The proximal exploration (PEX) and AdaLead results are replicated between the plots, since they are unaffected by choice of variational distribution.

Figure 2 shows the results for the fitness landscape experiments, and the BBO experimental results can be found in Figure 3. VSD is clearly the best performing method for all tasks, with the related method CbAS also performing well. We have consistently found the evolutionary-search based methods, PEX and AdaLead, to be effective on lower-dimensional problems, however we consistently observe their performance degrading as the dimension of the problem increases – e.g. on the BBO experiments. We suspect this is a direct consequence of their random mutation strategies being suited to exploration in low dimensions, but less efficient in higher dimensions compared to the learned generative models employed by VSD, CbAS, and DbAS. Our version of BORE (which is just the expected log-likelihood component of Equation 7) performs badly in most cases, and this is a direct consequence of its proposal distribution collapsing to a Kronecker delta centered on $\mathbf{x}_t^*$. In a non-batch setting, this behavior is not problematic, but shows how crucial the Kullback-Liebler (KL) divergence regularization of VSD is in this batch setting.

## 5   Conclusion

We have presented the problem of active generation (sequentially finding designs of a rare class under some experimental constraints), and a method for efficiently generating samples which we call variational search distributions (VSD). Underpinned by variational inference, VSD satisfies critical requirements and important desiderata, including learning generative models for feasible/fit sequences and batch candidate generation. We showcased the benefits of our method empirically on a set of combinatorially complete and high dimensional sequential-design biological problems and show that it can effectively learn powerful generative models of fit designs. There is a close connection between active generation and black box optimization, and with the advent of powerful generative models we hope that our explicit framing of generation of fit sequences would lead to further study of this connection. Finally, our framework can be generalized to more complex application scenarios, potentially involving other challenging combinatorial optimization problems [5], such as graph structures [3], and mixed discrete-continuous variables, which are worth investigating as future work directions.

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

## A    VSD as a Black-Box Optimization Lower Bound

A natural question to ask is how VSD relates to the BO objective for PI [14, Ch.7],

$$\mathbf{x}_t^* = \operatorname*{argmax}_{\mathbf{x}} \log \alpha_{PI}(\mathbf{x}, \mathcal{D}_N, \tau). \tag{8}$$

Firstly, we can see that the expected log-likelihood of term of Equation 3 lower-bounds this quantity.

**Proposition A.1.** *For a parametric model, $q(\mathbf{x}|\phi)$, given $\phi \in \Phi \subseteq \mathbb{R}^m$ and $q \in \mathcal{P} : \mathcal{X} \times \Phi \to [0, 1]$,*

$$\max_{\mathbf{x}} \log \alpha_{PI}(\mathbf{x}, \mathcal{D}_N, \tau) \geq \max_{\phi} \mathbb{E}_{q(\mathbf{x}|\phi)}[\log \alpha_{PI}(\mathbf{x}, \mathcal{D}_N, \tau)], \tag{9}$$

*and the bound becomes tight as $q(\mathbf{x}|\phi_t^*) \to \delta(\mathbf{x}_t^*)$, a Dirac delta function at the maximizer $\mathbf{x}_t^*$.*

Taking the argmax of the RHS will result in the variational distribution collapsing to a delta distribution at $\mathbf{x}_t^*$ for an appropriate choice of $q(\mathbf{x}|\phi)$. The intuition for Equation 9 is that the expected value of a random variable is always less than or equal to its maximum. The proof of this is in Daulton et al. [13], Staines & Barber [42]. Extending this lower bound, we can show the following.

**Proposition A.2.** *For a divergence $\mathbb{D} : \mathcal{P}(\mathcal{X}) \times \mathcal{P}(\mathcal{X}) \to [0, \infty)$, and a prior $p_0 \in \mathcal{P}(\mathcal{X})$,*

$$\max_{\mathbf{x}} \log \alpha_{PI}(\mathbf{x}, \mathcal{D}_N, \tau) \geq \max_{\phi} \mathbb{E}_{q(\mathbf{x}|\phi)}[\log \alpha_{PI}(\mathbf{x}, \mathcal{D}_N, \tau)] - \mathbb{D}[q(\mathbf{x}|\phi)\|p_0(\mathbf{x})]. \tag{10}$$

We can see that this bound is trivially true given the range of divergences, and this covers VSD as a special case. However, this bound is tight if and only if $p_0$ concentrates as a Dirac delta at $\mathbf{x}_t^*$ with an appropriate choice of $q(\mathbf{x}|\phi)$. In any case, the lower bound remains valid for any choice of informative prior $p_0$ or even a uninformed prior, which allows us to maintain the framework flexible to incorporate existing prior information whenever that is available.

## B    Related Work

We will consider related work first in terms of methods that have similar components to VSD, then second in terms of related problems to our specification of active generation. VSD can be viewed as one of many methods that makes use of the bound [42],

$$\max_{\mathbf{x}} f_{\bullet}(\mathbf{x}) \geq \max_{\phi} \mathbb{E}_{q(\mathbf{x}|\phi)}[f_{\bullet}(\mathbf{x})]. \tag{11}$$

The maximum is always greater than or equal to the expected value of a random variable. This bound is useful for black-box optimization (BBO) of $f_{\bullet}$, and becomes tight if $q(\mathbf{x}|\phi) \to \delta(\mathbf{x}^*)$, see Appendix A for more detail and VSD's relation to BO. Other well known methods that make use of this bound are evolution strategies (ES) and natural evolution strategies (NES) [52], variational

| Method | $w(\mathbf{x})$ | $\phi'$ | Fixed $\mathbf{x}^{(s)} \sim q(\mathbf{x}|\phi')$? |
|---|---|---|---|
| VSD | $\log \pi_{\theta*}(\mathbf{x}) + \log p(\mathbf{x}|\mathcal{D}_0) - \log q(\mathbf{x}|\phi)$ | $\phi$ | No |
| CbAS | $\pi_{\theta*}(\mathbf{x}) p(\mathbf{x}|\mathcal{D}_0) / q(\mathbf{x}|\phi_{t-1}^*)$ | $\phi_{t-1}^*$ | Yes |
| DbAS | $\pi_{\theta*}(\mathbf{x})$ | $\phi_{t-1}^*$ | Yes |
| BORE* | $\pi_{\theta*}(\mathbf{x})$ | $\phi$ | No |
| BOPR | $\alpha(\mathbf{x}, \mathcal{D}_N)$ | $\phi$ | No |

Table 1: How related methods can be adapted from Equation 12. VSD, CbAS and DbAS may also use a cumulative distribution representation of $\alpha_{\mathrm{PI}}(\mathbf{x}, \mathcal{D}_N, \tau)$ in place of $\pi_{\theta*}(\mathbf{x})$.

optimization (VO) [42, 6], estimation of distribution algorithms (EDA) [26], and BOPR [13]. For learning the parameters of the variational distribution, $\phi$, they variously make use of maximum likelihood estimation or the score function gradient estimator (REINFORCE) [53]. Algorithms that modify Equation 11 to stop the collapse of $q(\mathbf{x}|\phi)$ to a point mass for batch design include DbAS [9] and CbAS [8]. They use fixed samples $\mathbf{x}^{(s)}$ from $q(\mathbf{x}|\phi_{t-1}^*)$ for approximating the expectation, and then optimize $\phi$ using a weighted maximum-likelihood or variational style procedure. DbAS and CbAS were formulated for offline (non-sequential) tasks, they have often been used in a sequential setting. We can take a unifying view of many of these algorithms by recognizing the general gradient estimator, where we give each component in Table 1.

$$\mathbb{E}_{q(\mathbf{x}|\phi')}[w(\mathbf{x})\nabla_\phi \log q(\mathbf{x}|\phi)], \tag{12}$$

BORE* has been adapted to discrete $\mathcal{X}$ by using the score function gradient estimator and CbAS and DbAS have been adapted to use a CPE – their original derivations use the equivalent of a PI acquisition function.

A number of finite horizon methods have been applied to biological sequence BBO tasks, such as Amortized BO [46], GFlowNets [21], and the reinforcement learning based DynaPPO [2]. Latent space optimization (LSO)-like methods [17, 50, 43, 20] tackle optimization of sequences by encoding them into a continuous latent space within which candidate optimization or generation takes place. Selected candidates are decoded back into sequences before black box evaluation; see González-Duque et al. [18] for a comprehensive survey. VSD does not require a latent space nor an encoder, and as such can be seen as an amortized variant of probabilistic reparameterisation methods [13] or continuous relaxations [29]. Heuristic stochastic search methods such as AdaLead [39] and PEX [36] have also demonstrated strong empirical performance on these tasks. We compare the properties of the most relevant methods to our problem in Table 2.

In contrast to finding the maximum using BBO, active generation considers another problem – generating samples from a rare set of feasible solutions. Generation methods that estimate the super level-set distribution, $p(\mathbf{x}|y > \tau)$, include CbAS, which optimizes the forward KL divergence, $\mathbb{D}_{\mathrm{KL}}[p(\mathbf{x}|y > \tau)\|q(\mathbf{x}|\phi)]$ using importance weighted cross entropy estimation [37]. Batch-BORE [32] also optimizes the reverse KL divergence and uses CPE, but with Stein variational inference [28] for continuous and diverse batch candidates. There is a rich literature on the related task of active learning and BO for level-set estimation (LSE) [10, 19, 7, 54]. However, we focus on learning a generative model of a discrete space.

For active generation VSD, CbAS and DbAS all use an acquisition function defined in the *original* domain, $\mathcal{X}$, to weight gradients (see Equation 12) for learning a conditional generative model, from which $\mathbf{x}_{bt}$ are sampled. An alternative is to use *guided generation*, that is to train an unconditional generative model, and then have a discriminative model guide (condition) the samples from the unconditional model at test time. This plug-and-play of a discriminative model has shown promise for controlled image and text generation of pre-trained models [31, 12, 27, 55]. LaMBO [43] and LaMBO-2 [20] take a guided generation approach to solve the active generation problem. LaMBO uses an (unconditional) masked language model auto-encoder, and then optimizes sampling from its latent space using an acquisition function as a guide. LaMBO-2 takes a similar approach, but uses a diffusion process as the unconditional model, and modifies a Langevin sampling de-noising process with an acquisition function guide.

| Method | Rare $\mathbf{x} \in S$ (R1) | Sequential (R2) | Discrete $\mathcal{X}$ (R3) | Batch $\{\mathbf{x}_{bt}\}_{b=1}^{B}$ (R4) | Generative $q(\mathbf{x}\|\phi)$ (R5) | Guaranteed (D1) | Gradient descent (D2) | Scalable (D3) | General acq/reward fn. | Amortization |
|---|---|---|---|---|---|---|---|---|---|---|
| BOPR [13] | ✗ | ✓ | ✓ | ✗ | – | ✓ | ✓ | ✗ | ✓ | – |
| BORE [47] | ✗ | ✓ | – | ✗ | – | ✓ | ✓ | ✓ | ✗ | – |
| Batch BORE [32] | ✓ | ✓ | ✗ | ✓ | ✓ | ✓ | ✓ | ✓ | ✗ | ✓ |
| DbAS [9] | ✓ | ✓ | ✓ | ✓ | ✓ | ✗ | ✓ | ✓ | ✗ | ✓ |
| CbAS [8] | ✓ | ✓ | ✓ | ✓ | ✓ | ✗ | ✓ | ✓ | ✗ | ✓ |
| Amortized BO [46] | ✗ | ✓ | ✓ | ✓ | ✓ | ✗ | ✓ | ✓ | ✓ | ✓ |
| GFlowNets [21] | ✗ | ✓ | ✓ | ✓ | ✓ | ✗ | ✓ | ✓ | ✓ | ✓ |
| DynaPPO [2] | ✗ | ✓ | ✓ | ✓ | ✓ | ✗ | ✓ | – | ✓ | ✓ |
| AdaLead [39] | ✗ | ✓ | ✓ | ✓ | ✗ | ✗ | ✗ | – | ✗ | ✗ |
| PEX [36] | ✗ | ✓ | ✓ | ✓ | ✗ | ✗ | ✗ | – | ✗ | ✗ |
| GGS [24] | ✗ | ✗ | ✓ | ✓ | ✓ | ✗ | ✗ | ✗ | ✗ | ✗ |
| LSO e.g. [50] | ✗ | ✓ | ✓ | ✗ | ✓ | ✗ | ✓ | – | ✓ | – |
| LaMBO [43] | ✓ | ✓ | ✓ | ✓ | ✓ | ✗ | ✗ | – | ✓ | ✓ |
| LaMBO-2 [20] | ✓ | ✓ | ✓ | ✓ | ✓ | ✗ | ✓ | ✓ | ✓ | ✓ |
| VSD (ours) | ✓ | ✓ | ✓ | ✓ | ✓ | ✓ | ✓ | ✓ | ✗ | ✓ |

Table 2: Feature table of competing methods: ✓ has feature, ✗ does not have feature, – partially has feature, or requires only simple modification. We follow Swersky et al. [46] in their definition of amortization referring to the ability to use $q(\mathbf{x}\|\phi_{t-1}^{*})$ for warm-starting the optimization of $\phi_t$.

## C  Experiment Details

We compare our method, VSD, on a number of sequence design tasks and compare to existing baseline methods.

### C.1  Datasets

We use three well established datasets; a green fluorescent protein (GFP) from Aequorea victoria [38], an adeno-associated virus (AAV) [11]; and DNA binding activity to a human transcription factor (TFBIND8) [49, 4]. These datasets have been used variously by [9, 8, 1, 24, 21] among others. The GFP task is to maximize fluorescence, this protein consists of 238 amino acids, of which 237 can mutate. The AAV task us to maximize the genetic payload that can be delivered, and this protein has 28 amino acids, all of which can mutate. A complete combinatorial assessment is infeasible for these tasks, and so we use the convolution neural network oracle presented in [24] as *in-silico* ground truth. TFBIND8 contains a complete combinatorial enumeration of the effect of changing 8 nucleotides on binding to human transcription factor SIX6 REF R1 [4]. The dataset we use contains all 65536 sequences, prepared by [49]. We also use two datasets from recent works that enumerate the (near) complete combinatorial space of short sequences. The first dataset measures the antibiotic resistance of Escherichia coli metabolic gene folA, which encodes dihydrofolate reductase (DHFR) [33]. Only a sub-sequence of this gene is varied (9 nucleic acids which encode 3 amino acids), and so a near-complete (99.7%) combinatorial scan is available. For variants that have no fitness (resistance) data available, we give a score of -1. The next dataset is near-complete combinatorial scan of four interacting amino acid residues near the active site of the enzyme tryptophan synthase (TrpB) [23], with 159,129 unique sequences and fitness values, we use -0.2 for the missing fitness values. These residues are explicitly shown to exhibit epsistasis – or non-additive effects on catalytic function – which makes navigating this landscape a more interesting challenge from an optimization perspective. The properties of these datasets are summarized in Table 3.

### C.2  Evaluation

For all experiments we run a predetermined number of experimental rounds, $T = 10$, and we set the batch size to $B = 128$. In the first set of experiments, we use a fixed threshold, $\tau$, with the aim of estimating $p(\mathbf{x}|y > \tau)$ (or $S$ for non probabilistic models). For the next set of experiments, we set the threshold, $\tau$, adaptively for testing VSD's ability to find the fittest sequence.

| Dataset | $|\mathcal{V}|$ | $m$ | $|\mathcal{X}_{\text{available}}|$ | $|\mathcal{X}|$ |
|---|---|---|---|---|
| TFBIND8 | 4 | 8 | 65,536 | 65,536 |
| TrpB | 20 | 4 | 159,129 | 160,000 |
| DHFR | 4 | 9 | 261,333 | 262,144 |
| AAV | 20 | 28 | 42,340 | $20^{28}$ |
| GFP | 20 | 237 | 51,715 | $20^{237}$ |

Table 3: Vocabulary size, sequence length, and number of available sequences for each of the datasets we use in this work.

We compare against DbAS [9], CbAS [8], AdaLead [39], and PEX [36] – all of which we have adapted to use a CPE, BORE [47] that we have adapted to use the score function gradient estimator, and a naïve baseline that uses random samples from the prior, $p(\mathbf{x}|\mathcal{D}_0)$. To reduce confounding, all methods share the same surrogate model, acquisition functions, priors and variational distributions. For Adalead, we set $\kappa = 0.5$ since the CPE using $\tau$ is already performing the same thresholding.

## C.3 Fitness Landscapes

In this setting we fix $\tau$ over all rounds, for all competing methods, and we only consider the combinatorially (near) complete datasets. The primary measures by which we compare methods are precision, recall and performance (the last adapted from [21]),

$$\text{Precision}_t = \frac{1}{\min\{tB, |\mathcal{S}|\}} \sum_{r=1}^{t} \sum_{b=1}^{B} \mathbb{1}[y_{br} > \tau] \cdot \mathbb{1}[\mathbf{x}_{br} \notin \mathcal{X}_{b-1,r}^q], \tag{13}$$

$$\text{Recall}_t = \frac{1}{\min\{TB, |\mathcal{S}|\}} \sum_{r=1}^{t} \sum_{b=1}^{B} \mathbb{1}[y_{br} > \tau] \cdot \mathbb{1}[\mathbf{x}_{br} \notin \mathcal{X}_{b-1,r}^q], \tag{14}$$

$$\text{Performance}_t = \sum_{r=1}^{t} \sum_{b=1}^{B} y_{br} \cdot \mathbb{1}[\mathbf{x}_{br} \notin \mathcal{X}_{b-1,r}^q]. \tag{15}$$

Here $\mathcal{X}_{br}^q \subset \mathcal{X}$ is the set of experimentally queried sequences by the $b$th batch member of the $r$th round, including the initial training set. These are comparable among probabilistic and non probabilistic methods. Precision and recall measure the ability of a method to efficiently explore $\mathcal{S}$, where $\min\{tB, |\mathcal{S}|\}$ is the size of the selected set at round $t$ (bounded by the number of good solutions), and $\min\{TB, |\mathcal{S}|\}$ is the number of positive elements possible in the experimental budget. Strictly, recall should be normalized by $|\mathcal{S}|$, but we use $TB$ here since it may not be realistic to have the experimental budget to fully explore $\mathcal{S}$.

For the DHFR and TrpB experiments we set maximum fitness in the training dataset to be that of the wild type, and $\tau$ to be slightly below the wild type fitness value. We use a randomly selected $N_{\text{train}} = 2000$ below the wild-type fitness to initially train the CPE – which is a simple MLP (see Appendix C.7), we also explicitly include the wild-type. The thresholds and wild-type fitness values are; DHRF: $\tau = -0.1$, $y_{\text{wt}} = 0$, TrpB: $\tau = 0.35$, $y_{\text{wt}} = 0.409$. We follow the same procedure for the TFBIND8 experiment, however, there is no notion of a wild-type sequence in this data, and so we set $\tau = 0.75$, and $y_{\text{train max}} = 0.85$.

## C.4 Black Box Optimization

In this experiment we aim to find the global maximizers of the black box function, $(y^*, \mathbf{x}^*)$. For this, we set $\tau$ adaptively by specifying it as an empirical quantile of the observed target values,

$$\tau_t = \tilde{Q}_y^t(\gamma = p_{t-1}^\eta) \tag{16}$$

where $\tilde{Q}_y^t$ is the empirical quantile function of targets at round $t$, $p_{t-1}$ is a percentile from the previous round, and $\eta \in [0, 1]$ is a parameter that controls an 'annealing'-like schedule for $\tau_t$ that prioritizes exploration of the fitness landscape in earlier rounds and exploitation of known fit regions in later rounds. This is a strategy loosely-similar to [41]. The main measure of interest for these experiments is simple/instantaneous regret $r_t$ which quantifies how close the methods get to obtaining

the globally fittest sequence,

$$r_t = y^* - \max_y \{y_{bi}\}_{b=1,i=1}^{B,t}, \tag{17}$$

where $y^*$ is the fitness value of the maximum fitness sequence $\mathbf{x}^*$.

For these experiments we use the higher dimension AAV ($y^*$=19.54) and GFP ($y^*$=4.12) datasets to show that VSD can scale to higher dimensions. However, the $\mathcal{X}$ of these experiments is completely intractable to fully explore experimentally, and so we use a predictive oracle trained on all of the original experimental data as the ground-truth black-box function. This is the same strategy used in [8, 21, 48, 24] among others, and we use the exact CNN-based oracles from [24] for these experiments. Unfortunately, it has been shown that some of the oracles used in these experiments do not predict well out-of-distribution [45], limiting their applicability to real-world problems.

We follow [24] in the experimental settings for the AAV and GFP datasets, but we modify the maximum fitness training point and training dataset sizes to make them more amenable to a sequential optimization setting. The initial percentiles, schedule, and max training fitness values are; AAV: $p_0 = 0.8$, $\eta = 0.7$, $y_{max} = 5$, GFP: $p_0 = 0.8$, $\eta = 0.7$ $y_{max} = 1.9$. We again use a random $N_{train} = 2000$ for training the CPEs, which in this case are CNNs with an embedding layer input (same as the previous MLP), followed by two convolutional layers (with a kernel width of 7 residues) with max pooling, followed by two linear layers with leaky ReLU activations.

Again we find the simple proposal distribution in Equation 19 works as well as other more complex auto-regressive and transition, $q(\mathbf{x}_t|\mathbf{x}_{t-1}, \phi)$, proposal distributions. However, this time we make use of a prior distribution of the same form as Equation 19, fit using maximum likelihood on the same training sequences used for the CPE, regardless of fitness. We find that a uniform prior leads to far inferior convergence results for all methods (BORE and DbAS use this as the initial proposal distribution) apart from AdaLead and PEX, which use alternative generative heuristics. The Random method draws sequences randomly from this prior.

The results are summarized in Figure 3. VSD is among the leading methods, CbAS and DbAS for both experiments, but it is never significantly better. We can see that AdaLead, PEX and BORE all perform worse than random for reasons previously mentioned. Simple regret can drop below zero for these experiments since an oracle is used as the black box function, but the global maximizer is taken from the experimental data. This potentially highlights some of the overconfidence issues inherent in these oracles outlined in [45].

## C.5 Diversity Scores

The diversity of batches of candidates is a common thing to report in the literature, and to that end we present the diversity of our results here. We have taken the definition of diversity from [21] as,

$$\text{Diversity}_t = \frac{1}{B(B-1)} \sum_{\mathbf{x}_i \in \mathcal{D}_{Bt}} \sum_{\mathbf{x}_j \in \mathcal{D}_{Bt} \setminus \{\mathbf{x}_i\}} \text{Lev}(\mathbf{x}_i, \mathbf{x}_j), \tag{18}$$

where $\text{Lev} : \mathcal{X} \times \mathcal{X} \to \mathbb{N}_0$ is the Levenshtein distance. We caution the reader as to the interpretation of these results however, as more diverse batches often do not lead to better performance, precision, recall or simple regret (as can be seen from the Random method results). Though insufficient diversity can also explain poor performance, as in the case of BORE. Results for the fitness landscape experiment are presented in Figure 4, and black-box optimization for AAV & GFP in Figure 5.

## C.6 Prior and Variational Distributions

In this section we summarize the main variational distribution architectures considered for VSD, BORE, CbAS and DbAS, and the sampling distributions for the Random baseline method. Somewhat surprisingly, we find that we obtain consistently good results for the biological sequence experiments using a simple independent (or mean-field) variational distribution,

$$q(\mathbf{x}|\phi) = \prod_{m=1}^{M} \text{Categ}(x_m|\text{softmax}(\phi_m)), \tag{19}$$

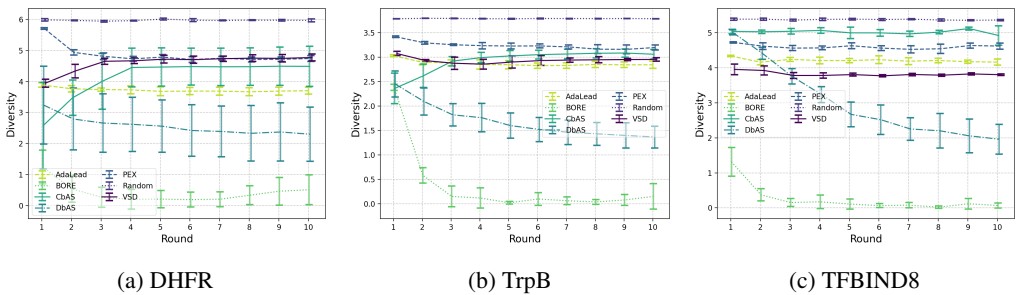

|(a) DHFR|(b) TrpB|(c) TFBIND8|

Figure 4: Fitness landscape diversity results. Higher is more diverse, as defined by Equation 18.

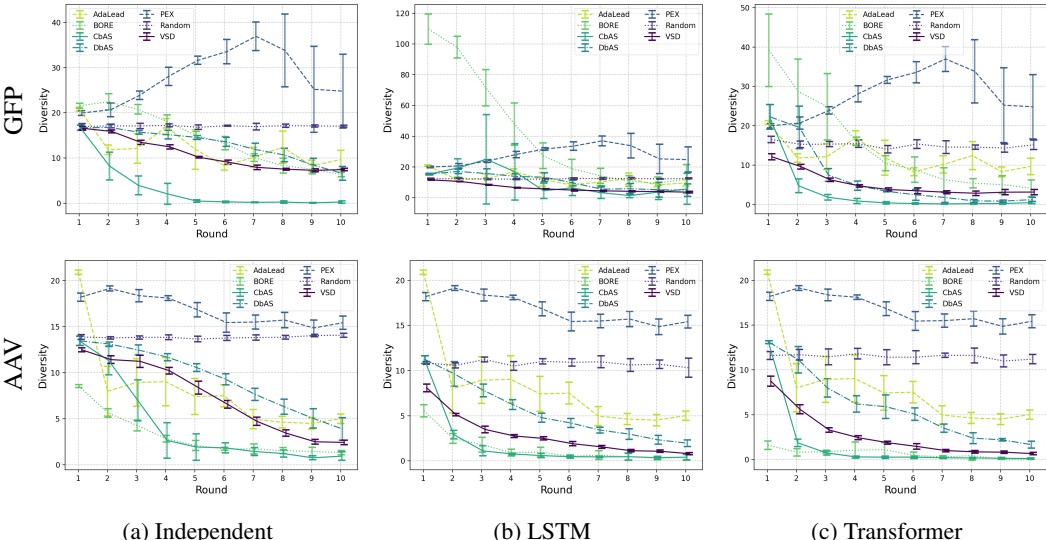

(a) Independent          (b) LSTM          (c) Transformer

Figure 5: Black-box optimization results for diversity on GFP and AAV with independent and auto-regressive variational distributions. Higher is more diverse, as defined by Equation 18. The PEX and AdaLead results are replicated between the plots, since they are unaffected by choice of variational distribution.

where $x_m \in \mathcal{V}$ and $\phi_m \in \mathbb{R}^{|\mathcal{V}|}$. However, this simple mean-field distribution was not capable of generating convincing handwritten digits. We have also tested a variety of transition variational distributions,

$$q(\mathbf{x}_t|\mathbf{x}_{t-1}, \phi) = \prod_{m=1}^{M} \mathrm{Categ}(x_{tm}|\mathrm{softmax}(\mathrm{NN}_m(\mathbf{x}_{t-1}, \phi))), \qquad (20)$$

where $\mathrm{NN}_m(\mathbf{x}_{t-1}, \phi)$ is the $m^{\mathrm{th}}$ vector output of a neural network that takes a sequence from the previous round, $\mathbf{x}_{t-1}$, as input. We have implemented multiple neural net encoder/decoder architectures for $\mathrm{NN}_m(\mathbf{x}_{t-1}, \phi)$, but we did not consider architectures of the form $\mathrm{NN}_m(\phi)$ since the variational distribution in Equation 19 can always learn a $\phi_m = \mathrm{NN}_m(\phi')$. We found that none of these transition architectures significantly outperformed the mean-field distribution (Equation 19) when it was initialized well (e.g. fit to the CPE training sequences). We also implemented auto-regressive variational distributions of the form,

$$q(\mathbf{x}|\phi) = \mathrm{Categ}(x_1|\mathrm{softmax}(\phi_1)) \prod_{m=2}^{M} q(x_m|x_{1:m-1}, \phi_{1:m}) \quad \text{where,} \qquad (21)$$

$$q(x_m|x_{1:m-1}, \phi_{1:m}) = \begin{cases} \mathrm{Categ}(x_m|\mathrm{softmax}(\mathrm{LSTM}(x_{m-1}, \phi_{m-1:m}))), \\ \mathrm{Categ}(x_m|\mathrm{softmax}(\mathrm{DTransformer}(x_{1:m-1}, \phi_{1:m}))). \end{cases}$$

For a LSTM recurrent neural network (RNN) and a decoder-only transformer with a causal mask, for the latter see Phuong & Hutter [34, Algorithm 10 & Algorithm 14] for maximum likelihood training and sampling implementation details respectively. We list the configurations of the LSTM and transformer variational distributions in Table 4. We use additive positional encoding for all of these models.

| | Configuration | AAV | GFP |
|---|---|---|---|
| LSTM | Layers | 4 | 4 |
| | Network size | 32 | 32 |
| | Embedding size | 10 | 10 |
| Transformer | Layers | 1 | 1 |
| | Network Size | 64 | 64 |
| | Attention heads | 2 | 2 |
| | Embedding size | 20 | 20 |

Table 4: LSTM and transformer network configuration.

### C.7 Class Probability Estimator Architectures

For the fitness landscape experiments on the smaller combinatorially complete datasets we use a two-hidden layer MLP, with an input embedding layer. The architecture is given in Figure 6 (a). For the larger dimensional AAV and GFP datasets and Ehrlich function benchmark, we use the convolutional architecture given in Figure 6 (b). On all but the Ehrlich benchmark, five fold cross validation was used to select the hyper parameters before the CPEs are trained on the whole training set for use in the subsequent experimental rounds. Model updates are performed by retraining on the whole query set.

## D The VSD Algorithm

The VSD algorithm is summarized in Algorithm 1.

---

**Algorithm 1** VSD optimization loop with CPE.

---

**Require:** Threshold $\gamma_1$ and $f_\tau$, dataset $\mathcal{D}_N$, black-box $f_\bullet$, prior $p(\mathbf{x}|\mathcal{D}_0)$, CPE $\pi_\theta(\mathbf{x})$, variational family $q(\mathbf{x}|\phi)$, budget $T$ and $B$.
1: **function** FITMODELS($\mathcal{D}_N, \tau$)
2:      $\mathcal{D}_N^z \leftarrow \{(z_n, \mathbf{x}_n)\}_{n=1}^N$, where $z_n = \mathbb{1}[y_n > \tau]$
3:      $\theta^* \leftarrow \text{argmin}_\theta \mathcal{L}_{\text{CPE}}(\theta, \mathcal{D}_N^z)$
4:      $\phi^* \leftarrow \text{argmax}_\phi \mathcal{L}_{\text{ELBO}}(\phi, \theta^*)$
5:      **return** $\phi^*, \theta^*$
6: **for** round $t \in \{1, \ldots, T\}$ **do**
7:      $\tau_t \leftarrow f_\tau(\{y : y \in \mathcal{D}_N\}, \gamma_t)$
8:      $\phi_t^*, \theta_t^* \leftarrow$ FITMODELS($\mathcal{D}_N, \tau_t$)
9:      $\{\mathbf{x}_{bt}\}_{b=1}^B \leftarrow q(\mathbf{x}|\phi_t^*)$
10:     $\{y_{bt}\}_{b=1}^B \leftarrow \{f_\bullet(\mathbf{x}_{bt}) + \epsilon_{bt}\}_{b=1}^B$
11:     $\mathcal{D}_{N+B} \leftarrow \mathcal{D}_N \cup \{(\mathbf{x}_{bt}, y_{bt})\}_{b=1}^B$
12: $\tau_* \leftarrow f_\tau(\{y : y \in \mathcal{D}_N\}, \gamma_*)$
13: $\phi^*, \theta^* \leftarrow$ FITMODELS($\mathcal{D}_N, \tau_*$)
14: **return** $\phi^*, \theta^*$

---

```
Sequential(
    Embedding(
        num_embeddings=A,
        embedding_dim=8
    ),
    Dropout(p=0.2),
    Flatten(),
    LeakyReLU(),
    Linear(
        in_features=8 * M,
        out_features=32
    ),
    LeakyReLU(),
    Linear(
        in_features=32,
        out_features=1
    ),
)
```

(a) MLP architecture

```
Sequential(
    Embedding(
        num_embeddings=A,
        embedding_dim=10
    ),
    Dropout(p=0.2),
    Conv1d(
        in_channels=10,
        out_channels=16,
        kernel_size=7,
    ),
    LeakyReLU(),
    MaxPool1d(
        kernel_size=2 or 4,
        stride=2 or 4,
    ),
    Conv1d(
        in_channels=16,
        out_channels=16,
        kernel_size=7,
    ),
    LeakyReLU(),
    MaxPool1d(
        kernel_size=2 or 4,
        stride=2 or 4,
    ),
    Flatten(),
    LazyLinear(
        out_features=128
    ),
    LeakyReLU(),
    Linear(
        in_features=128,
        out_features=1
    ),
)
```

(b) CNN architecture

Figure 6: CPE architectures used for the experiments in PyTorch syntax. $A = |\mathcal{V}|$, $M = M$, GFP uses a max pooling kernel size and stride of 4, all other datasets and benchmarks use 2.

