# OpenReview forum: "Variational Search Distributions"
_NeurIPS.cc/2024/Workshop/BDU — NeurIPS BDU Workshop 2024 Oral_

### Official Review · Reviewer_ZQa4 · 2024-09-25
**Variational inference for active search**

**Rating:** 9
**Confidence:** 3

**Review:**

This paper addresses the problem of active search (e.g., find compounds that would be useful as pharmaceuticals) by formulating the problem as finding an approximate posterior distribution over desirable designs given some prior over designs. The authors frame their method as variational inference and call it Variational Search Distributions (VSD). The exposition of the methods is clear, including the use of a score function gradient estimator and class probability estimation. VSD appears to generalize several related optimization algorithms and convincing experimental results are shown.

Questions for the authors to consider:

1.	How critical is the choice of prior distribution over designs? How might this depend on the dimension of the search space?
2.	Can you give an idea of the data necessary to either train a reasonable surrogate model or CPE and have the method succeed?

---

### Official Review · Reviewer_xvp9 · 2024-09-26
**Clear paper with a novel algorithm for variational search**

**Rating:** 8
**Confidence:** 3

**Review:**

# Summary

This paper proposes a family of variational search problems, outlines their requirements and desiderata, and proposes an algorithm that satisfies them. By search problems, the authors refer to estimating densities of the sort $p(\boldsymbol{x} | y > \tau)$. Having access to said distributions is vital, and could conduce to generalizations of Bayesian optimization.

The authors estimate said probability of improvement using a variational approach, which they name Variational Search Distributions (VSD): approximate the posterior using a parametrized proposal distribution $q(\boldsymbol{x}|\phi)$, finding the parameters by maximizing the evidence lower bound. The authors derive a simple version of this objective, which can be readily optimized using out-of-the-box gradient methods. The method is presented as scalable, diverse, and generative.

# Review

The paper is a pleasure to read. The problem is well motivated, the proposed method is clear, and the derivations seem sound. Furthermore, the problem is relevant, and approached from a practitioner’s perspective. Sequence design indeed requires the left side of Table 1. The method is tested thoroughly against baselines.

With this in mind, I would like to recommend for clear acceptance.

Let me finish with some general comments and questions for improvement/discussion in future work:

- One detail in which I am not fully convinced is on whether (R4) is satisfied w. diversity: Could you/did you measure **the diversity** of the samples from the current variational approximation?
- There is no justification for the choice of batch size. One easy way to choose a batch size would be to approach practitioners and *ask*. Several of these analyses are done in plates of size ~98. Ideally, one would also see a sensitivity analysis of this parameter in the experimentation.
- When it comes to incorporating prior knowledge through $p(\boldsymbol{x})$. How difficult would it be to use log-likelihoods coming out of protein LLMs like the ESM family? My intuition says it should work out-of-the-box, but I might be mistaken. An experiment incorporating these likelihoods instead of a uniform prior would be pretty convincing, and I expect it to speed-up convergence significantly.

One final small comment: wich → which in the conclusions.

---

### Decision · Program_Chairs · 2024-10-09

Accept (Oral)